# Carboranes as Lewis Acids: Tetrel Bonding in CB$_{11}$H$_{11}$ Carbonium Ylide

**Maxime Ferrer [1], Ibon Alkorta [1], José Elguero [1] and Josep M. Oliva-Enrich [2,***

[1]  Instituto de Química Médica (CSIC), Juan de la Cierva, 3, E-28006 Madrid, Spain; maxime.ferrer@iqm.csic.es (M.F.); ibon@iqm.csic.es (I.A.); iqmbe17@iqm.csic.es (J.E.)
[2]  Instituto de Química-Física "Rocasolano" (CSIC), Serrano, 119, E-28006 Madrid, Spain
*  Correspondence: j.m.oliva@iqfr.csic.es; Tel.: +34-915619400

**Abstract:** High-level quantum-chemical computations (G4MP2) are carried out in the study of complexes featuring tetrel bonding between the carbon atom in the carbenoid CB$_{11}$H$_{11}$—obtained by hydride removal in the C-H bond of the known *closo*-monocarbadodecaborate anion CB$_{11}$H$_{12}^{(-)}$ and acting as Lewis acid (LA)—and Lewis bases (LB) of different type; the electron donor groups in the tetrel bond feature carbon, nitrogen, oxygen, fluorine, silicon, phosphorus, sulfur, and chlorine atomic centres in neutral molecules as well as anions H$^{(-)}$, OH$^{(-)}$, and F$^{(-)}$. The empty radial $2p_r$ vacant orbital on the carbon centre in CB$_{11}$H$_{11}$, which corresponds to the LUMO, acts as a Lewis acid or electron attractor, as shown by the molecular electrostatic potential (MEP) and electron localization function (ELF). The thermochemistry and topological analysis of the complexes {CB$_{11}$H$_{11}$:LB} are comprehensively analysed and classified according to shared or closed-shell interactions. ELF analysis shows that the tetrel C$\cdots$X bond ranges from very polarised bonds, as in H$_{11}$B$_{11}$C:F$^{(-)}$ to very weak interactions as in H$_{11}$B$_{11}$C$\cdots$FH and H$_{11}$B$_{11}$C$\cdots$O=C=O.

**Keywords:** Lewis acid; carborane; carbonium ylide; tetrel bond; quantum chemistry; electron density; ELF





## 1. Introduction

The very stable B$_{12}$H$_{12}^{(2-)}$ dianion and its neutral dicarbon counterparts *ortho*-(1,2-C$_2$B$_{10}$H$_{12}$), *meta*-(1,7-C$_2$B$_{10}$H$_{12}$), and *para*-carborane (1,12-C$_2$B$_{10}$H$_{12}$) are icosahedral systems that are closely related to elemental boron. Their isoelectronic analogue, *closo*-monocarbadodecaborate anion CB$_{11}$H$_{12}^{(-)}$, first prepared in 1967 [1] and further with other synthesis methods [2,3], is similarly resistant to cage degradation, and many derivatives have been synthesized as described in the literature [4]. The stability and three-dimensional aromaticity of CB$_{11}$H$_{12}^{(-)}$ has also been explained using quantum-chemical computations [5]. Extraction of hydride H$^{(-)}$ in the C-H bond from CB$_{11}$H$_{12}^{(-)}$ leads to a carbocation ylide or carbenoid (**1**) with a vacant radial $2p_r$ orbital on the cage carbon atom as shown in Figure 1. On the other hand, reaction mechanisms of polyhedral (car)boranes and their derivatives are scarce in the literature and further research is needed in this respect [6–8]. Thus, the permethylated carbenoid analogue CB$_{11}$Me$_{11}$ has been postulated as a reaction intermediate during the extraction of the L substituent from L-CB$_{11}$Me$_{11}$ carboranes (L = BrCH$_2$CH$_2$ or (CF$_3$)$_2$CHO) by electrophiles [6,7], further reacting with arenes in the presence of (CF$_3$)$_2$CHOH to generate 1-aryl-CB$_{11}$Me$_{11}$ products [6–8]. The methyl groups in the permethylated anion CB$_{11}$Me$_{12}^{(-)}$ have substantial CH$_3^{(-)}$ (methide) character according to DFT computations [6–8] and can easily bind to transition metal and main group elements.

On the other hand, in recent years, tetrel bonding—defined as an interaction between any electron donating system (ED) and a group 14 element acting as Lewis acid—has called the attention of both experimentalists [9] and theoreticians [10–12]. Here the carbon centre

in (**1**) is clearly an acceptor of electrons or Lewis acid; hence, we can define a tetrel bonding interaction with an electron donor (ED) or Lewis base, as shown in Figure 1c.

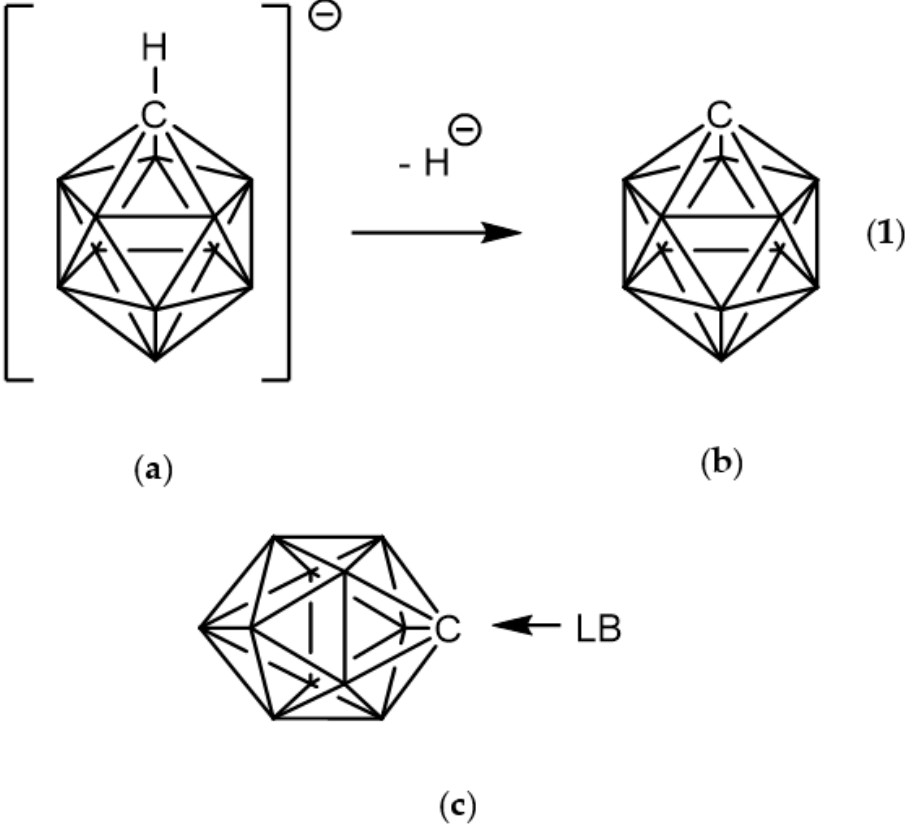

**Figure 1.** Removal of hydride from the C-H bond in (**a**) *closo*-monocarbadodecaborate anion $CB_{11}H_{12}^{(-)}$ leads to (**b**) carbocation ylide or carbenoid $CB_{11}H_{11}$ (**1**). (**c**) Complex formation between (**1**) and a Lewis base (LB). All vertices correspond to B-H moieties except for the carbon vertex.

The goal of this work is to study the electronic interaction between the naked carbon vertex in the carbenoid (**1**) with a series of electron donor molecules and anions leading to tetrel C-X bonds. The chosen 18 LB systems, including the anions $H^{(-)}$, $F^{(-)}$, and $OH^{(-)}$, are displayed below in Scheme 1 with the corresponding label.

| $H^{(-)}$ | $CH_2$ | $CF_2$ | $C\equiv O$ | $N_2$ | $NH_3$ |
|---|---|---|---|---|---|
| (**2**) | (**3**) | (**4**) | (**5**) | (**6**) | (**7**) |

| $NH=CH_2$ | $N\equiv CH$ | $OH^{(-)}$ | $OH_2$ | $O=CH_2$ | $O=C=O$ |
|---|---|---|---|---|---|
| (**8**) | (**9**) | (**10**) | (**11**) | (**12**) | (**13**) |

| $F^{(-)}$ | FH | $SiH_2$ | $PH_3$ | $SH_2$ | ClH |
|---|---|---|---|---|---|
| (**14**) | (**15**) | (**16**) | (**17**) | (**18**) | (**19**) |

**Scheme 1.** The chosen set of molecules acting as Lewis base (LB) and forming a tetrel bond with the C atom from carbenoid (**1**) according to Figure 1c.

## 2. Computational Methods

Electronic structure quantum-chemical computations were carried out using the G4MP2 model [13], which is a fourth-generation method available in the Gaussian16 scientific software [14]. This method combines density-functional theory [15,16] and second-order perturbation theory [17] and provides an accurate and economical method

for thermochemical predictions. The G4(MP2) model works as follows: The geometries of the molecules are optimized at the B3LYP/6-31G(2df,p) level of theory, and then a series of single point energy calculations at higher levels of theory are computed. The zero-point energy, E(ZPE), is based on B3LYP/6-31G(2df,p) frequencies scaled by 0.9854, the same as in G4 theory. The first energy calculation is at the triples-augmented coupled cluster level of theory, CCSD(T), with the 6-31G(d) basis set, i.e., CCSD(T)/6-31G(d). This energy is then modified by a series of energy corrections to obtain a total energy $E_0$. For more details on the G4(MP2) method, the reader is referred to Reference [13]. In the particular case of the **1**:LB complexes, we computed the enthalpy and free energy differences between the complex and separated systems **1** and LB at room temperature as indication of stability of the complex. All complexes included in this work correspond to energy minimum structures, checked with frequency computations. The quantum theory of atoms-in-molecules (QTAIM) [18,19] was used in the topological analysis of the electron density of the **1**:LB complexes with the scientific software AIMAll [20]. This method is based on the analysis of the electron density $\rho$, its gradient $\nabla\rho$, and the corresponding Laplacian $\nabla^2\rho$. For further aspects of this methodology, the reader is referred to the above References [17,18] and Section 3.3.1 below. The electron localisation function (ELF) [21,22] was also used in the topological analysis of the complexes. The ELF is a distribution function which measures the probability of finding two electrons with the same spin, as further described in Section 3.3.2 below. The TopMod09 package [23] was used for the ELF calculations.

## 3. Results

### 3.1. Geometries of Complexes (1:n), n = 2–19

In Figure 2a,b we display the G4MP2 optimized geometries of $CB_{11}H_{12}^{(-)}$ with $C_{5v}$ symmetry—the (**1:2**) complex and the carbonium ylide $CB_{11}H_{11}$ (**1**), with the different tetrel complexes (**1:n**), **n = 2–19** and structural parameters when necessary, in order to highlight the atomic rearrangements undergone due to the complexation process. The loss of $H^{(-)}$ in the C-H leads also to a structure with $C_{5v}$ symmetry—with a geometrical change which involves a considerable flattening of the $CB_5$ pentagonal pyramid with expansion of the corresponding $B_5$ pentagon, since there is an increase of the B-B bond distance of $\Delta = +0.022$ Å. As we proceed down the cage from the top (C atom), the B-B bond differences are +0.022 Å, +0.022 Å, and −0.015 Å. Quite noticeable is the shortening of the apical intracage $C\cdots B$ distance ($\Delta\sim-0.3$ Å). The B-H bond distances hardly change at all, with a very slight shortening upon loss of $H^{(-)}$ with $\Delta = -0.007$ Å, −0.008 Å, and −0.001 Å, from top to bottom, respectively. If we take the C and B cage nuclei as point charges and define a distorted $C_{5v}$ icosahedron, the corresponding volumes are $V(CB_{11}H_{12}^{(-)}) = 12.03$ Å$^3$ and $V(CB_{11}H_{11}) = 11.90$ Å$^3$, and therefore there is a shrinkage of the cage by $\Delta V = -0.13$ Å$^3$ (1%). In summary, the extraction of hydride in the C-H bond from $CB_{11}H_{12}^{(-)}$ implies a minor change in the cage volume with a flattening of the top $CB_5$ pentagonal pyramid and minor changes as we proceed down the cage from the top C atom. In Figure 2c–s, we display the optimised geometries for the remaining complexes with the coordinates gathered in the Supplementary Material (SI, Tables S1–S10). The shortest and longest $C\cdots X$ tetrel interactions correspond to the original anion $CB_{11}H_{12}^{(-)}$ or complex (**1:2**) and the $CB_{11}H_{11}\cdots O=C=O$ tetrel complex (**1:13**), respectively. In the latter case, the interaction is clearly non-covalent in origin with $d(C\cdots O) = 2.693$ Å, as will be discussed later on. We should also emphasize the $C\cdots X$ interaction of the $CH_2$ and $SiH_2$ complexes (**1:3**) and (**1:16**), respectively, with the LB groups tilted from the $C_5$ axis of rotation. In all other systems, including the $CF_2$ complex (**1:4**), the $C\cdots X$ bond is aligned with the $C_5$ axis of rotation.

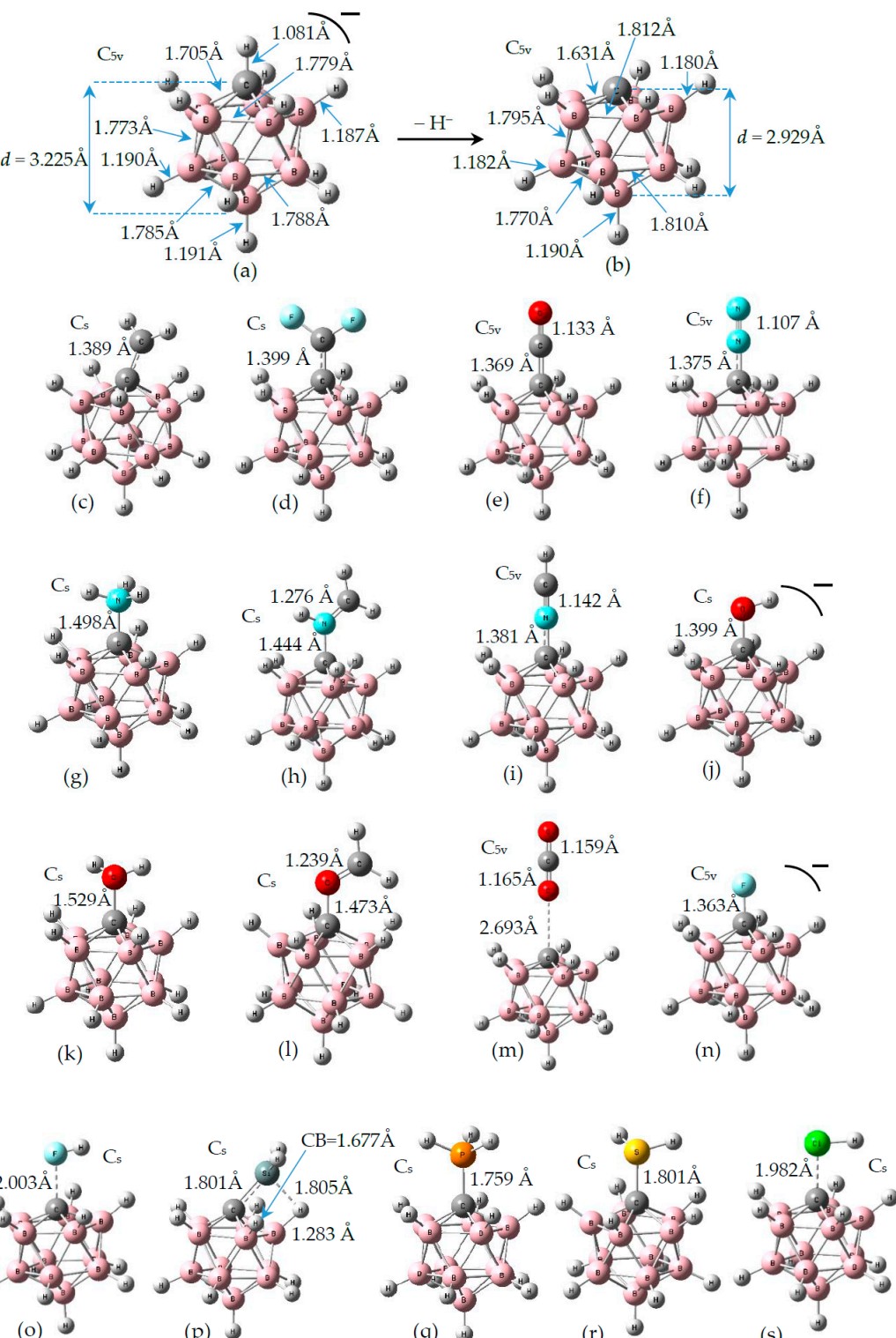

**Figure 2.** Structures of the G4MP2 optimized geometries for the 18 tetrel complexes (**1**:*n*)—with *n* = 2–19—considered in this work, following Scheme 1: (**a**) The known anion $CB_{11}H_{12}^{(-)}$ corresponding to complex (**1:2**), (**b**) carbenoid $CB_{11}H_{11}$ (**1**), (**c**) (**1:3**), (**d**) (**1:4**), (**e**) (**1:5**), (**f**) (**1:6**), (**g**) (**1:7**), (**h**) (**1:8**), (**i**) (**1:9**), (**j**) (**1:10**), (**k**) (**1:11**), (**l**) (**1:12**), (**m**) (**1:13**), (**n**) (**1:14**), (**o**) (**1:15**), (**p**) (**1:16**), (**q**) (**1:17**), (**r**) (**1:18**), and (**s**) (**1:19**).

In Figure 3, we display the d(C···X) distances in the tetrel bonding complexes, ordered from shortest to longest. Clearly, we can classify five groups according to the C···X distances, in increasing order: (i) The original anion $CB_{11}H_{12}^{(-)}$ or (**1:2**) complex; (ii) com-

plexes with d$\sim$1.4–1.5 Å including complexes (**1:k$_2$**), with k$_2$ = (3–12, 14); (iii) complexes with d$\sim$1.8 Å, including complexes (**1:k$_3$**), with k$_3$ = 16–18; (iv) complexes with d$\sim$2.0 Å, including complexes (**1:15**) and (**1:19**); and finally (v) the (**1:13**) complex with d$\sim$2.7 Å.

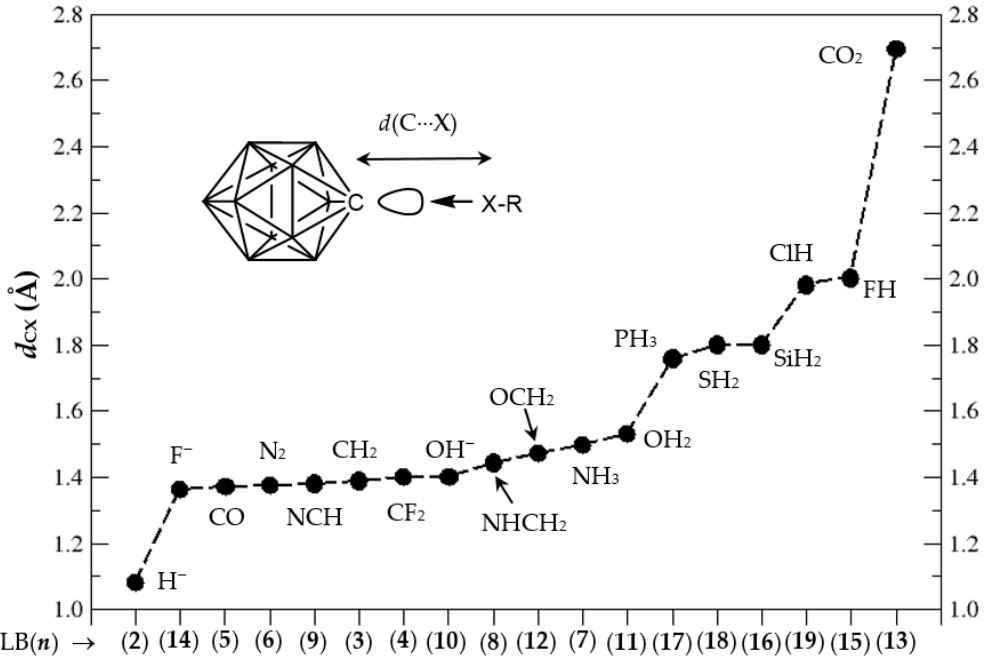

**Figure 3.** C$\cdots$X distances d$_{CX}$ (Å) in the tetrel complexes (**1:n**), *n* = **2–19**, from Figure 2, in increasing order; x axis corresponds to Lewis base (LB) *n* and y axis to d$_{CX}$ distances (Å), respectively.

### 3.2. Thermochemistry of Complexes (1:n), n = 2–19

In this subsection, we predict the ΔH and ΔG of the tetrel bonding complexes (**1:n**), *n* = **2–19**. In Table 1, we gather the computed enthalpies and free energies at the G4MP2 level of theory.

**Table 1.** Enthalpy (ΔH) and free energy (ΔG) of formation for complexes (**1:n**), *n* = **2–19**, in kJ·mol$^{-1}$. LB: Lewis base.

|  | *n* = 2 | 3 | 4 | 5 | 6 | 7 | 8 | 9 | 10 |
|---|---|---|---|---|---|---|---|---|---|
| **LB** | H$^{(-)}$ | CH$_2$ | CF$_2$ | C≡O | N$_2$ | NH$_3$ | NH=CH$_2$ | N≡CH | OH$^{(-)}$ |
| **ΔH** | −858.6 | −504.9 | −326.2 | −211.1 | −50.5 | −268.6 | −294.6 | −170.6 | −659.3 |
| **ΔG** | −822.3 | −457.7 | −279.7 | −162.9 | −2.8 | −226.8 | −242.7 | −120.8 | −618.0 |

|  | 11 | 12 | 13 | 14 | 15 | 16 | 17 | 18 | 19 |
|---|---|---|---|---|---|---|---|---|---|
| **LB** | OH$_2$ | O=CH$_2$ | O=C=O | F$^{(-)}$ | FH | SiH$_2$ | PH$_3$ | SH$_2$ | ClH |
| **ΔH** | −116.2 | −139.8 | −14.3 | −571.5 | −0.7 | −418.0 | −280.9 | −186.1 | −47.6 |
| **ΔG** | −72.1 | −87.9 | 25.7 | −529.7 | 31.3 | −366.8 | −238.1 | −140.9 | −8.9 |

The free energy of formation for complexes (**1:n**) is always negative except for (**1:13**) and (**1:15**) complexes, namely the O=C=O and FH complexes, respectively. Small negative values ($|\Delta G| < 10$ kJ·mol$^{-1}$) are obtained for complexes (**1:6**) and (**1:19**) with Lewis bases N$_2$ and ClH, respectively. Therefore, all complexes with negative ΔG should be formed at room temperature spontaneously, provided an isolated Lewis acid (**1**) approaches an isolated Lewis base. On the other hand, the enthalpies of formation are negative for all complexes, an indication that the bond energies of a given complex (**1:n**) have a lower value than the bond energies of separated systems (**1**) and (**n**). In order to better visualize

the similarities and differences of the thermochemical aspects of complexes (**1**:*n*), we plot ΔG and ΔH vs. *n* in increasing order of each state function.

As shown in Figure 4 from left to right in the abscissa, the ΔG (black) and ΔH (blue) in complex formation follow the same order as function of Lewis base number except for ClH (**1:19**) and $N_2$ (**1:6**), where the order is inverted in the ΔH tendency as compared to ΔG. Hence, the formation of anionic complexes are the most energetic and favourable ones in the order (**1:2**), (**1:10**), (**1:14**) corresponding to Lewis bases $H^{(-)}$, $OH^{(-)}$, and $F^{(-)}$, respectively; then follow complexes (**1:3**), (**1:16**), and (**1:4**) corresponding to Lewis bases $CH_2$, $SiH_2$ and $CF_2$, respectively, namely the carbene series. A plateau with complexes (**1:8**), (**1:17**), and (**1:7**) follows with Lewis bases $NH=CH_2$, $PH_3$, and $NH_3$, respectively. A smaller (positive) slope of ΔG/ΔH vs. *n* appears with complexes (**1:5**), (**1:18**), (**1:9**), (**1:12**), and (**1:11**) always in increasing order, which correspond to Lewis bases $C≡O$, $SH_2$, $N≡CH$, $O=CH_2$, and $OH_2$, respectively. The weakest bound complexes with ΔG < 0 correspond to (**1:19**) and (**1:6**) with Lewis bases ClH and $N_2$, respectively. Finally, complexes (**1:13**) and (**1:15**) with Lewis bases O=C=O and FH, respectively, show a predicted quantum-chemical value of ΔG > 0, and therefore one should not expect a spontaneous formation of these complexes at room temperature. It is noteworthy to mention the tiny value ΔH(**1:15**) = −0.7 kJ·mol⁻¹ for FH attachment to (**1**); this number is within the accuracy of the method and therefore a heat of formation for complex (**1:15**) or the bond energy on both sides of the equation remains unaltered.

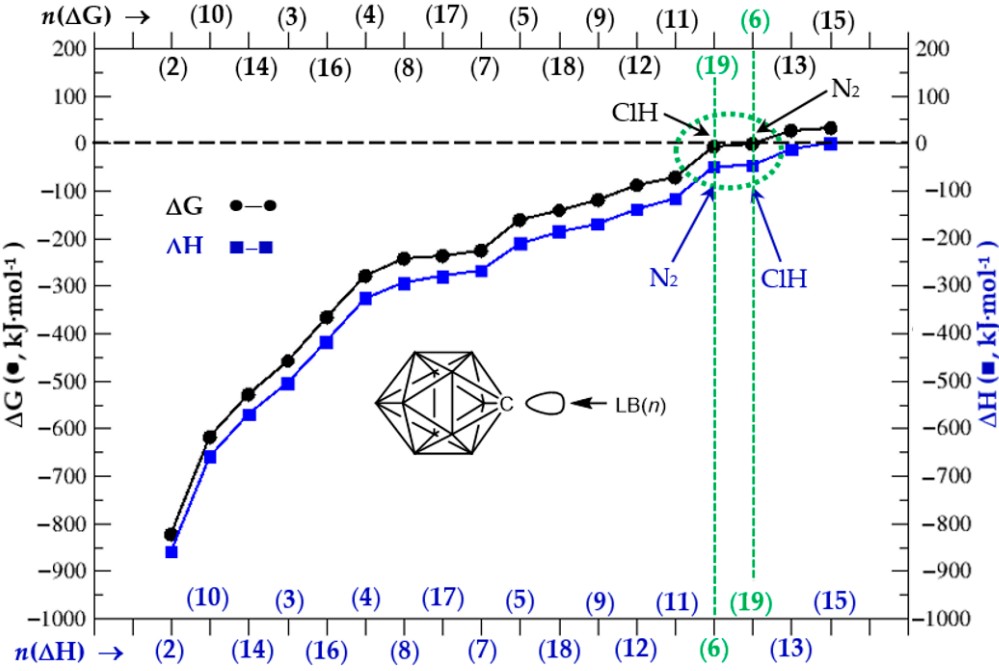

**Figure 4.** ΔG and ΔH vs. Lewis base *n* in respective increasing order, all in kJ·mol⁻¹.

### 3.3. Electronic Structure of Complexes (1:n), n = 2–19

In Figure 5, we show for (**1**) the molecular electrostatic potential (MEP) and the electron localization function (ELF). These electronic structure features are computed using the optimized geometry of the system with the G4MP2 method—B3LYP/6-31G(2df,p) model chemistry for structure optimization. As noticed in Figure 5a, the shape of the MEP and the corresponding π-hole just on top of the C ylide centre shows the electron-attraction nature of this region of the molecule. In the ELF from Figure 5b, we show disynaptic V(B,H) yellow basins corresponding to the B-H bonds; the ELF distribution around the $CB_{11}$ icosahedral cage can be partitioned into green disynaptic and trisynaptic basins, as we will describe below in Section 3.3.2.

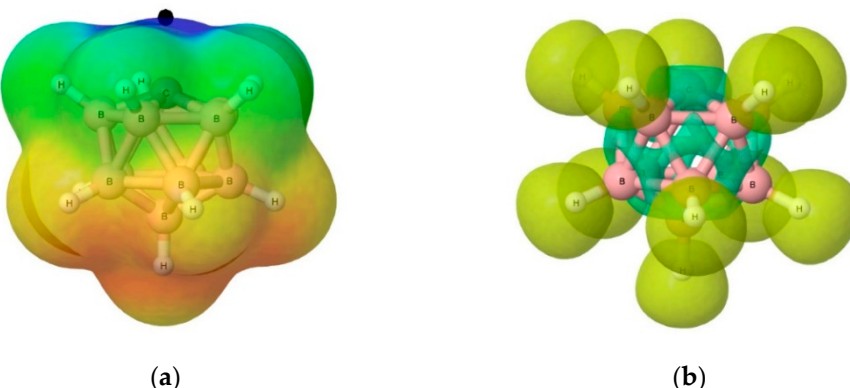

**Figure 5.** Electronic structure of carbonium ylide (**1**) $CB_{11}H_{11}$: (**a**) Molecular electrostatic potential $V(\mathbf{r})$. Red colour for $V(\mathbf{r}) < -0.015$ au, Blue colour for $V(\mathbf{r}) > 0.03$ au. The black dot indicates the localization of the $\pi$-hole (0.061 au), and (**b**) Electron Localization Function (ELF) with an isosurface of ELF = 0.75. Computations with the G4MP2 level of theory.

The molecular electrostatic potential (MEP) is the potential energy of a proton at a particular location near a molecule. Negative electrostatic potential corresponds to an attraction of the proton by the concentrated electron density in the molecules. The MEP of (**1**)—Figure 5b—shows that the potential energy of a proton is most positive above the C atom with a $\pi$-hole of +0.061 au, hence a repulsive region for a proton approaching (**1**), or electron acceptor region. The MEP is smoothly changing from positive to negative values of the potential energy as the proton moves from the C atom down to the B skeleton cage region. A proton would then be attached more favourably to the lower region of the carbonium ylide (**1**). In other words, Lewis bases, electron donors, and nucleophiles should then tend to bind through the C atom of the ylide, hence the study of the tetrel bonding in the complexes (**1**:*n*).

### 3.3.1. Atoms-in-Molecules (AIM) Topological Analysis of Complexes (**1**:*n*), *n* = **2–19**

The Quantum Theory of Atoms in Molecules (QTAIM) [18,19] is a useful tool for analysing the electronic structure of a polyatomic many-electron system, with the electron density $\rho(\mathbf{r})$ as the central function. The topological properties of $\rho(\mathbf{r})$ are analysed with the gradient of $\nabla\rho(\mathbf{r})$, the Laplacian of $\rho(\mathbf{r})$, $\nabla^2\rho(\mathbf{r})$, and the eigenvalues of the Hessian matrix of the electron density $\lambda_1, \lambda_2, \lambda_3$. The critical points are those with $\vec{\nabla}\rho = \vec{0}$ and a bond critical point (BCP) has $\lambda_3 > 0$ associated with the bond path direction, and $\lambda_1 < 0$, $\lambda_2 < 0$ the two latter associated to two directions where $\nabla^2\rho$ is a maximum; the BCP $(-,-,+)$ appears at the intersection of the bond path with the interatomic surface S. Other critical points are classified according to the signs of $\lambda_i$: Nuclei positions with $(-, -, -)$; ring critical points with $(-,+,+)$; cage critical points with $(+,+,+)$. We should also introduce the local electron kinetic (G > 0), potential (V < 0), and total (H) energy densities, with H = G + V, also useful parameters at the BCP for the description of the type of bonding interaction between atoms in a many-electron system [24]. In the SI (Table S11), we provide the computed values of $\rho(\mathbf{r})$, $\nabla\rho(\mathbf{r})$, $\nabla^2\rho(\mathbf{r})$, G, V, and H for the BCP found between the X atom of the Lewis base in contact with the C ylide centre in (**1**), for all complexes (**1**:*n*), *n* = **2–19**.

In Figure 6a, we plot the electron density at the BCP for the C$\cdots$X interaction vs. $d(CX)$ distance. The largest values of $\rho_{BCP}$ correspond to $CH_2$ ($\rho_{BCP}(CH_2) = 0.32\ e/a_0{}^3$), $CF_2$, and CO, followed by $H^{(-)}$, $OH^{(-)}$, $NHCH_2$, $N_2$, and $F^{(-)}$. Another group follows with lower values, $NH_3$, $OCH_2$, $PH_3$, $OH_2$, $SH_2$, and further down, $SiH_2$ and ClH with similar values. Finally, the lowest $\rho_{BCP}$ correspond to FH and $CO_2$, the latter with $\rho_{BCP}(CO_2) = 0.01\ e/a_0{}^3$. We should emphasize that the ratio $\rho_{BCP,max}(CH_2)/\rho_{BCP,min}(CO_2) = 32$ gives an idea of the topological differences in these BCPs. Given the different type of C$\cdots$X interactions in the complexes, the $\rho_{BCP}$ vs. $d(CX)$ can be fit to an approximate negative exponential curve with $\rho_{BCP}(d_{CX}) = a + b\cdot\exp(-c\cdot d_{CX})$, with $a = -0.022$, $b = +3.222$, and $c = -1.751$, and a

correlation factor of $R^2 = 0.99$ for closed-shell interacting complexes: $CO_2$, FH, $SiH_2$, $N_2$, and NCH. This curve is displayed in the SI (Figure S1). In general, very good correlations appear if we fix the two interacting atoms both belonging to the same row of the Periodic Table [25–27].

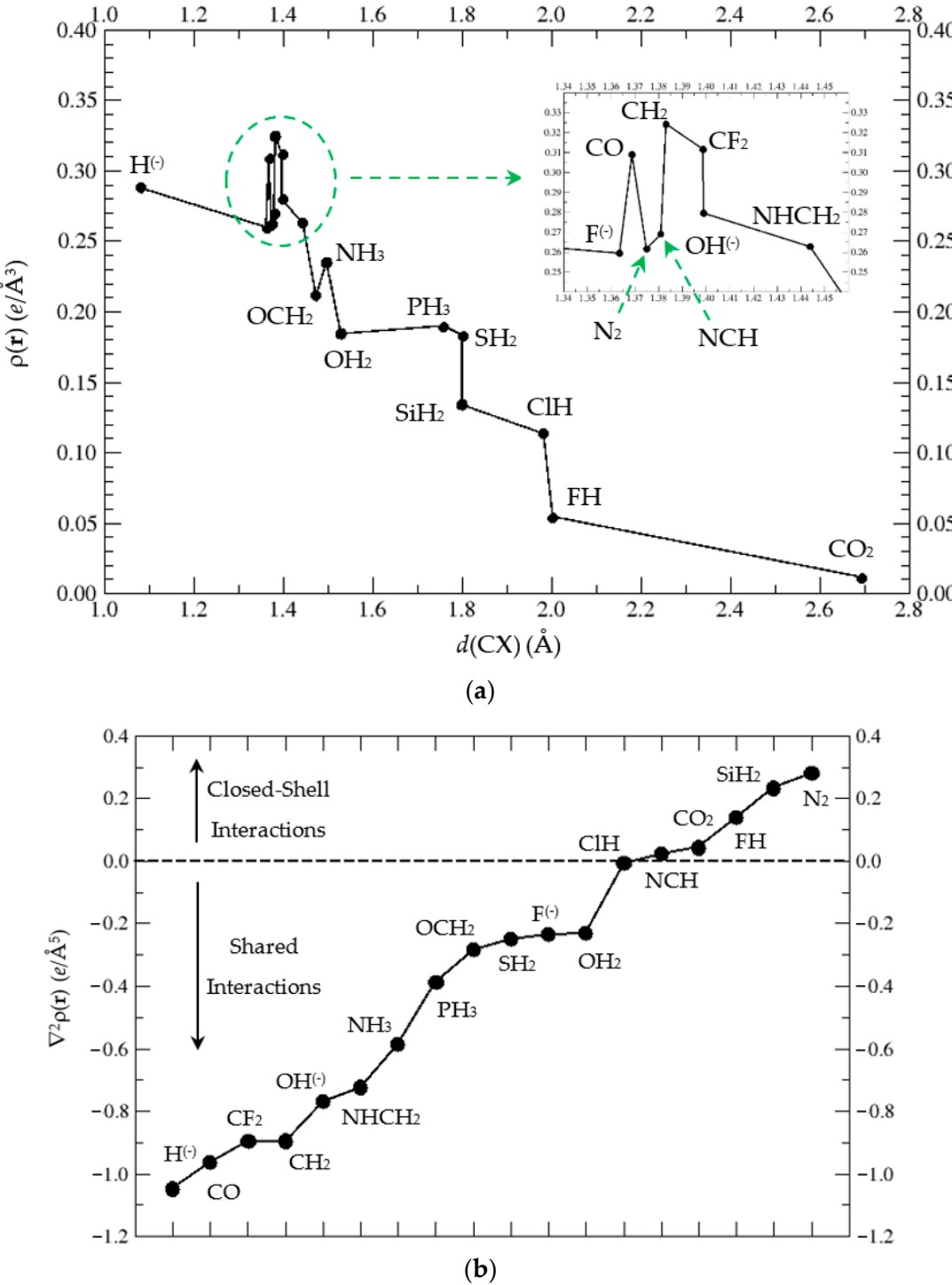

**Figure 6.** *Cont.*

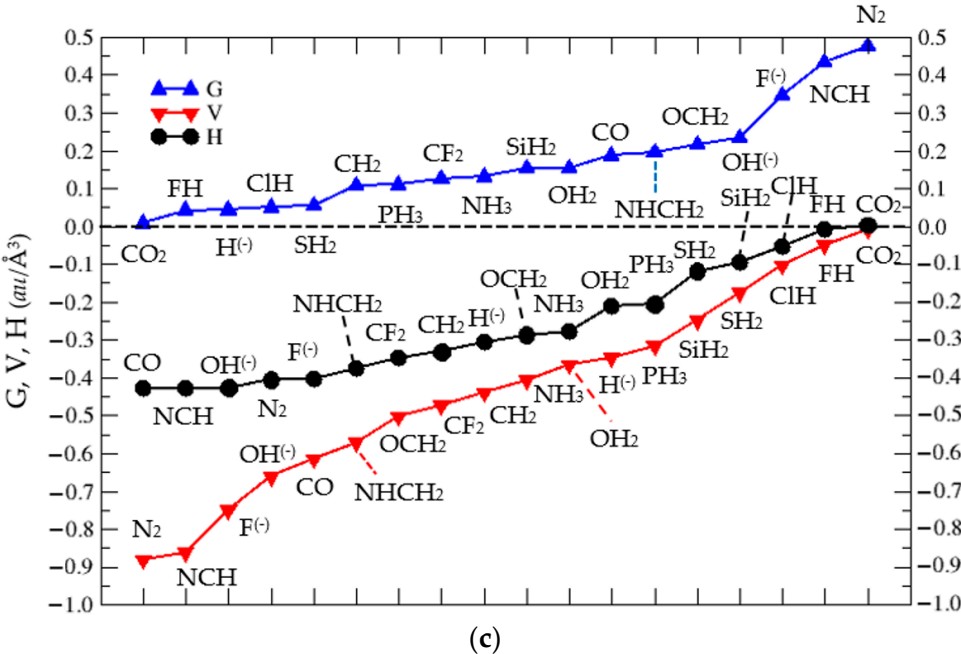

**Figure 6.** (**a**). $\rho(\mathbf{r})$ at the BCP of the C$\cdots$X interaction in complexes (**1**:*n*) vs. *d*(CX), with *n* = **2–19**. We label each point with the corresponding LB. (**b**) $\nabla^2\rho(\mathbf{r})$ at the C$\cdots$X BCP for complexes (**1**:*n*), *n* = **2–19**. (**c**) Plot of G (▲), V (▼), and H (●) vs. Lewis base LB(*n*) in increasing order of G, V, and H, respectively, at the Bond Critical Point (BCP) in the C$\cdots$X interaction of complexes (**1**:*n*), *n* = **2–19**, as described in Scheme 1.

In order to estimate the type of interaction we need to go beyond the electron density at the BCP and analyse the second derivative, the Laplacian $\nabla^2\rho$, and the kinetic, potential, and total energy, G, V, and H respectively, of the BCPs in complexes (**1**:*n*). In Figure 6b, the Laplacian is plotted vs. LB(*n*) in increasing order. Clearly, we can distinguish the shared interactions for $\nabla^2\rho < 0$ in the lower left corner and closed-shell interactions for $\nabla^2\rho > 0$ in the upper right corner of Figure 6b. The two-electron sharing in complexes with $H^{(-)}$, CO, $CF_2$, and $CH_2$ is large, and it is diminished up to $OH_2$. For the complex with HCl, $\nabla^2\rho = -0.0067\ e/\text{Å}^5$, namely in the limit between shared and closed-shell interactions. For positive Laplacians, in increasing order, the LB in the (**1**:*n*) complexes correspond to: NCH, $CO_2$, FH, $SiH_2$, and $N_2$; in these systems, the closed-shell interactions are important.

A further analysis of the BCP in the C$\cdots$X interactions of the (**1**:*n*) complexes can be found in the values of G, V, and H—with H = G + V being the total energy—as displayed in Figure 6c. The kinetic energy G is associated with repulsion in the bonding region, and the local potential energy density or local virial field V is a measure of the average effective potential field experienced by a single electron in a many-particle system. Thus, according to Figure 6c, the G and V profiles are inverted for $N_2$, NCH, $F^{(-)}$, and $OH^{(-)}$, but due to the nature of different nuclei in the C-X interactions, this is not always the case, as seen when we follow the profiles as function of LB(*n*).

### 3.3.2. Electron Localisation Function (ELF) Analysis of Complexes (**1**:*n*), *n* = **2–19**

We should emphasize that ELF is a function which reports the probability of finding an electron pair with opposite spins in a region of space. Using a certain isovalue, we are able to define regions of space, basins, with a certain probability to find an electron pair. For example, in the plots of the ELF, we used an isovalue of 0.83; in other words, we plot regions of space where we have a high probability to find a pair of electrons. Once the basins are defined, we can integrate the electronic density into those basins, which are the values reported above in Table 2 and correspond to the number of electrons in that basin. In ELF analysis, the partition of space is not based on the electron density, as in AIM, but on the ELF probability function. In order to better understand from the electronic

structure point of view the tetrel bonding in the (**1**:*n*) complexes, we further computed the electron localisation function (ELF) [21,22,28], a measure of the likelihood of finding an electron in the neighborhood space of a reference electron located at a given point and with the same spin; therefore, ELF is a measure of the Pauli repulsion or exchange interaction [29,30]. The ELF for the carbonium ylidene $CB_{11}H_{11}$ (**1**) is shown in Figure 4. ELF values ranges from zero to one (normalized and without units). In the SI file we provide the ELF for all complexes (**1**:*n*) not shown here (Table S12), and below we have selected four cases with short, medium, long, and very long $C\cdots X$ distances, according to Figure 3 above: (i) LB = $H^{(-)}$, complex (**1**:**2**), (ii) LB = $N_2$, complex (**1**:**6**), (iii) LB = $PH_3$, complex (**1**:**17**), and (iv) LB = $CO_2$, complex (**1**:**13**). In Table 2, we gather the ELF function for these four complexes.

**Table 2.** Electron localisation function (ELF) (isovalue 0.83 au) for complexes $H_{11}B_{11}C:H^{(-)} \rightarrow$ (**1**:**2**), $H_{11}B_{11}C:N\equiv N \rightarrow$ (**1**:**6**), $H_{11}B_{11}C:PH_3 \rightarrow$ (**1**:**17**), and $H_{11}B_{11}C:O=C=O \rightarrow$ (**1**:**13**). Basin labels are depicted for each complex. Population of ELF disynaptic basins V(C, X) in bold.

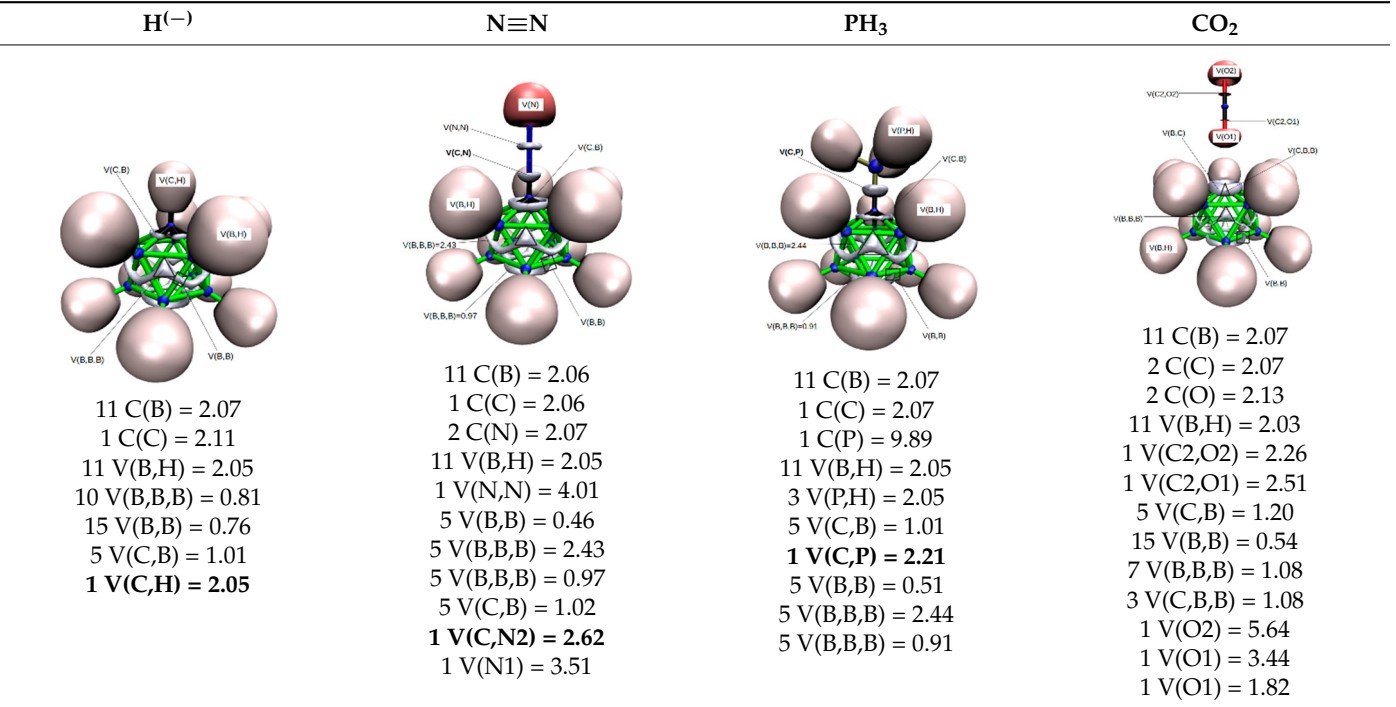

| $H^{(-)}$ | $N\equiv N$ | $PH_3$ | $CO_2$ |
|---|---|---|---|
| 11 C(B) = 2.07 | 11 C(B) = 2.06 | 11 C(B) = 2.07 | 11 C(B) = 2.07 |
| 1 C(C) = 2.11 | 1 C(C) = 2.06 | 1 C(C) = 2.07 | 2 C(C) = 2.07 |
| 11 V(B,H) = 2.05 | 2 C(N) = 2.07 | 1 C(P) = 9.89 | 2 C(O) = 2.13 |
| 10 V(B,B,B) = 0.81 | 11 V(B,H) = 2.05 | 11 V(B,H) = 2.05 | 11 V(B,H) = 2.03 |
| 15 V(B,B) = 0.76 | 1 V(N,N) = 4.01 | 3 V(P,H) = 2.05 | 1 V(C2,O2) = 2.26 |
| 5 V(C,B) = 1.01 | 5 V(B,B) = 0.46 | 5 V(C,B) = 1.01 | 1 V(C2,O1) = 2.51 |
| **1 V(C,H) = 2.05** | 5 V(B,B,B) = 2.43 | **1 V(C,P) = 2.21** | 5 V(C,B) = 1.20 |
| | 5 V(B,B,B) = 0.97 | 5 V(B,B) = 0.51 | 15 V(B,B) = 0.54 |
| | 5 V(C,B) = 1.02 | 5 V(B,B,B) = 2.44 | 7 V(B,B,B) = 1.08 |
| | **1 V(C,N2) = 2.62** | 5 V(B,B,B) = 0.91 | 3 V(C,B,B) = 1.08 |
| | 1 V(N1) = 3.51 | | 1 V(O2) = 5.64 |
| | | | 1 V(O1) = 3.44 |
| | | | 1 V(O1) = 1.82 |

Below each ELF function of a given complex (**1**:*n*), we also report the function value and average population for the different types of basins. A threshold of 0.2 electrons is considered as to include or not a basin in a group. Below each ELF function of a given complex (**1**:*n*), we report the population of the different basins. In order to avoid dealing with a long list of populations and after observing that basins involving the same elements have similar populations, we decided to report only for each type of basins the average population. A threshold of 0.2 electrons was chosen to decide if two basins belong to the same group or not. For instance, if we consider basins $V(B_1,B_2)$ and $V(B_1,B_3)$ with populations of 0.8 and 1.5 electrons, respectively, they belong to different groups. In bold letters, we report the value of the disynaptic basin corresponding to the tetrel $C\cdots X$ interaction.

In Table 2 we use the following notation:

- C: core basin
- V(X): monosynaptic basin, which can be associated to a lone pair
- V(X,X): disynaptic basin
- V(X,X,X): trisynaptic basin

- 5 V(B,B) = 0.42: There are 5 disynaptic basins involving two boron atoms with an average population of 0.42 electrons.

  In the ELF representation, the following colours are used:

- Blue: core basin
- Red: monosynaptic basin
- Grey: polysynaptic basin
- Beige: V(X,H) basin
- Atoms: Boron (green), Carbon (black), Hydrogen (silver), Oxygen (red), and Phosphorus (tan).

Thus, according to Table 2, in the complex (**1:2**), the existing anion $CB_{11}H_{12}^{(-)}$, the 11 core electrons from the boron cage, the $1s^2(B)$ ones, are gathered in the C(B) core basins, which have an average population of 2.07 electrons; the same description applies for the single C atom and the C(C) = 2.11 basin. The 11 B-H bonds on each non-naked B vertex of the icosahedron correspond to the 11 V(B,H) = 2.05, basically a two-electron covalent B-H bond. Then the distribution of the remaining valence cage electrons (corresponding to 2$s$ and 2$p$ electrons from B and C, plus the surplus electron or negative charge of the anion) are distributed in the 10 V(B,B,B) trisynaptic basins with an average population of 0.81 electrons, the 15 V(B,B) disynaptic basins with a population of 0.76 electrons, and the 5 V(C,B) disynaptic basins with ∼1 electron in them. The V(C,H) disynaptic basin, with a population of 2.05, corresponds to the C···X = C-H bond in (**1:2**), a two-centre two-electron bond. Addition of the values of all basins leads to the number of electrons in $CB_{11}H_{12}^{(-)}$: 74.

As reported in Table 2, the C···X interaction in the selected complexes can be described by the presence or absence of V(C,X) valence basins and its population. For the complexes (**1:6**) and (**1:17**), these values are $V_{(1:6)}(C,N) = 2.62$ and $V_{(1:17)}(C,P) = 2.21$, respectively. Therefore, ELF describes the C···X for the $N_2$ complex as a bond, with a multiplicity close to 1.5, between the C(ylide) and N nuclei and for the $PH_3$ complex a C(ylide)P single bond, with additional 0.2 electrons. As regard to the (**1:13**) complex with $CO_2$, the ELF does not localize a basin between the C(ylide) and the O=C=O molecule; the absence of ELF basins indicate that electron pairs are not shared, and therefore, the interaction is not covalent. However, other interactions such as ionic or non-covalent are possible even in the absence of ELF basins. The lone pairs from $N_2$ and $CO_2$ appear as red monosynaptic basins, as displayed in Table 2.

## 4. Discussion

The presence of a filled or empty lone pair on the C atom in the known anion $CB_{11}H_{12}^{(-)}$, complex (**1:2**), depends on whether we remove a proton or a hydride from the C-H bond, leading to a dianion $[CB_{11}H_{11}]^{(2-)}$ (**1b**) or a carbonium ylide $CB_{11}H_{11}$ (**1**), respectively; the latter process is shown in Figure 1. The tetrel complexes (**1:**$n$) presented in the previous section show a rich variety of thermochemical and electronic structure features with tetrel C···X interactions from different nuclei: X = {H, C, N, O, F, Si, P, Cl}. The C(ylide) centre in (**1**) confers to this particular molecule with a Lewis acid (LA) character, hence the tetrel denomination. The strength for electronic attachment in (**1**) is given by the computed free energy of formation (**1**) + ($n$) → (**1:**$n$). The strongest complexes correspond to those formed with anions $H^{(-)}$, $OH^{(-)}$, and $F^{(-)}$, and the weakest complexes to those formed with FH, $CO_2$, $N_2$, and ClH. The C···X distance varies considerably in all complexes, ranging from 1.081 Å for (**1:2**), LB = $H^{(-)}$, to 2.694 Å for (**1:13**), LB = $CO_2$. The complex strength is not related to the C···X distance; namely, complexes with similar C···X distances may have different free energy of formation, e.g., the free energy of formation for complexes (**1:14**) LB = $F^{(-)}$ and (**1:6**) LB = $N_2$ is −530 kJ·$mol^{-1}$ and −3 kJ·$mol^{-1}$, respectively, with very similar d(C-X) distances, 1.364 Å and 1.375 Å.

Examples of recent related systems is the 3D analogue of phenyllithium, the lithiacarborane $CB_{11}H_{11}:Li^{(-)}$, studied in solution as a solid and by quantum-chemical computations [31]. Indeed, $Li^{(-)}$ is a very poor Lewis base but certainly attaches to (**1**), as

recently shown, and defined as the lithiated mono-anion $[\text{Li}-\text{CB}_{11}\text{H}_{11}]^{(-)}$. On the other hand, this process can also be seen as a carborane dianion $[\text{CB}_{11}\text{H}_{11}]^{(2-)}$—a very reactive species—attached to $\text{Li}^{(+)}$, as described in Reference [31]. Table 8-1 from the book by Grimes, *Carboranes* [4], reports hundreds of compounds derived from $\text{CB}_{11}\text{H}_{12}^{(-)}$, and therefore this is a rich field not only from a synthetic point of view but also for studying the electronic structure of tetrel $\text{C}\cdots\text{X}$ bonds in these compounds, and especially if the isolated tetrel complexes (**1**:*n*) could ever be synthesized, taking into account that this work is purely theoretical with predictive quantum-chemical computations.

The electronic structure of the complexes has been analysed thoroughly with AIM and ELF methods, showing the $\text{C}\cdots\text{X}$ sharing and closed-shell interactions in the complexes according to the values of the Laplacian of the electron density. In Table 3, we gather the ELF values for disynaptic basins V(C, X) in the $\text{C}\cdots\text{X}$ region showing values of ELF: we can find very polarised C-F bonds in (1:14)—only one electron in the $\text{C}\cdots\text{X}$ region—single C-X bonds for $\text{H}^{(-)}$ and $\text{NH}=\text{CH}_2$ and intermediate cases, such as in complex (1:5) with a 1.5 multiplicity C-C bond for the CO complex. No V(C,X) disynaptic basins are found for $\text{CO}_2$ and FH, an indication of the poor electron-donating ability of these Lewis bases (LB) with indeed long $d$(C-X) distances and positive free energies of formation $\Delta\text{G} > 0$, hence confirming the unlikely formation of these two complexes.

**Table 3.** Population of ELF disynaptic basins V(C, X) in tetrel complexes (**1**:*n*), *n* = **2–19**, describing the $\text{C}\cdots\text{X}$ interaction. LB = Lewis base.

| *n* | LB | V(C, X) | *n* | LB | V(C, X) |
|---|---|---|---|---|---|
| 2 | $\text{H}^{(-)}$ | 2.05 | 11 | $\text{OH}_2$ | 1.56 |
| 3 | $\text{CH}_2$ | 2.53 | 12 | $\text{O}=\text{CH}_2$ | 1.65 |
| 4 | $\text{CF}_2$ | 2.71 | 13 | $\text{O}=\text{C}=\text{O}$ | - |
| 5 | $\text{C}\equiv\text{O}$ | 2.78 | 14 | $\text{F}^{(-)}$ | 0.99 |
| 6 | $\text{N}\equiv\text{N}$ | 2.62 | 15 | FH | - |
| 7 | $\text{NH}_3$ | 1.76 | 16 | $\text{SiH}_2$ | 2.40 |
| 8 | $\text{NH}=\text{CH}_2$ | 2.02 | 17 | $\text{PH}_3$ | 2.21 |
| 9 | $\text{N}\equiv\text{CH}$ | 2.34 | 18 | $\text{SH}_2$ | 1.85 |
| 10 | $\text{OH}^{(-)}$ | 1.32 | 19 | ClH | 1.11 |

According to the Cambridge structural database(CSD) [32], several tetrel complexes derived from (1) have been characterised [33–39] where the C(ylide) centre interacts with a nitrogen atom from neutral aminoderivatives including pyridine. These structures are shown in the SI file as Table S13. The shortest $\text{C(ylide)}\cdots\text{N}$ distance, $d(\text{C}\cdots\text{N}) = 1.477$ Å, corresponds to the pyridine complex 1-(4-methoxypyridinium)-1-carba-*closo*-dodecaborane [33]. The longest $\text{C(ylide)}\cdots\text{N}$ distance, $d(\text{C}\cdots\text{N}) = 1.554$ Å, corresponds to the complex 12-iodo-1-(4-pentylquinuclidine)-1-carba-*closo*-dodecaborane [34]. There is a tetrel complex of (**1**) with $\text{NH}_3$, 1-amino-2-fluorocarba-*closo*-dodecaborane [35], where one B-H vertex hydrogen atom on position 2 has been substituted by a fluorine atom with $d(\text{C}\cdots\text{N}) = 1.486$ Å. Our (**1:7**) tetrel complex $\text{H}_{11}\text{B}_{11}\text{C} \leftarrow :\text{NH}_3$ has a predicted $d(\text{C}\cdots\text{N}) = 1.498$ Å according to the G4MP2 computational model.

## 5. Conclusions

The results presented in this work show that by means of quantum-chemical computations we should expect the formation of tetrel complexes between the icosahedral carbonium ylide $\text{CB}_{11}\text{H}_{11}$—derived from extraction of $\text{H}^{(-)}$ in the known anion $\text{CB}_{11}\text{H}_{12}^{(-)}$—and a set of simple molecules and anions. The driving force of formation for these complexes can be accounted for from thermochemical quantum-chemical computations using statistical mechanics implemented in the scientific software Gaussian16 [14], and the results indicate that all the complexes should be formed with the exception of the FH and $\text{CO}_2$ molecules, with $\text{N}_2$ and ClH complexes with indeed very low, though negative, free energies of formation.

The tetrel C$\cdots$X interactions in all complexes have been thoroughly studied by means of AIM and ELF methods, hence defining the type of bond and interaction, ranging from very polarised bonds, with one electron in the C$\cdots$X moiety, to intermediate cases as in the carbenes $CH_2$ and $CF_2$ and silane $SiH_2$, with one and a half electrons in the C$\cdots$X region.

The existence of known tetrel complexes of the carbonium ylide $CB_{11}H_{11}$ with amino derivatives, including pyridine, opens the door toward further experimental and theoretical studies in the electronic structure of unusual bonds and interactions between C(ylide) centres in carboranes and other atoms.

We hope that the results from this work can be used for the isolation of reactive species, such as the recently found dianion derived from proton extraction in the well-known carborane anion $CB_{11}H_{12}{}^{(-)}$, a key molecule in the description of 3D aromaticity within boron chemistry.

**Supplementary Materials:** The following are available online at https://www.mdpi.com/article/10.3390/cryst11040391/s1, Tables S1–S10. G4MP2 optimised geometries of complexes (**1**:**n**). Table S11. AIM data for complexes (**1**:**n**). Table S12. ELF data for complexes (**1**:**n**) not displayed in the main text.

**Author Contributions:** Writing-original draft preparation, computations, and data curation: J.M.O.-E., M.F., I.A., and J.E. All authors have read and agreed to the published version of the manuscript.

**Funding:** This research was funded by Spanish MICINN, grant number CTQ2018-094644-B-C22 and Comunidad de Madrid, grant number P2018/EMT-4329 AIRTEC-CM.

**Data Availability Statement:** Not applicable.

**Conflicts of Interest:** The authors declare no conflict of interest.

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
