# Peer review of "Carboranes as Lewis Acids: Tetrel Bonding in CB11H11 Carbonium Ylide"

_crystals, doi:10.3390/cryst11040391_

Round 1
Reviewer 1 Report
The authors of the article entitled Tetrel bonding in CB11H11 carbonium ylide show an in-depth analysis of the chemical interaction between a hefty number of Lewis bases and the CB11H11- anion. They use a high-accurate method as the G4MP2 model to compute the thermodynamics of the bond formation reaction. Then, the authors use real-space approaches to determine the nature of the chemical bonding between the Lewis bases and the carbonium ylide of the CB11H11- anion. The authors conclude that all the bases studied can form spontaneously tetryl bonds with the carbonium ylide at room temperature, but FH and CO2. They also find that N2 and HCl show virtually zero free energy of formation.
The article is well structured and the methods have been wisely selected. Although some of the statements made by the authors are not supported by the results, the main message of the article could be appealing to the community. Therefore, I do recommend this article for publication after minor revision.
Comments
- Page 3; Line 100. The authors do not cite the code used to perform the ELF analysis. The references relative to ELF, 21 and 22, are not correct. They refer to the local analysis of the electron density.
- Page 3; Line 102. The definition of ELF is wrong. ELF is connected to the probability of finding two electrons with the same spin.
- Page 7; Line 271. It is written ... the LUMO of the carbonium ylide has considerable wave function amplitude around the C atom ... Strictly speaking, for a many-electron system the LUMO is not a wave function. Additionally, this sentence is referred to as a Kohn-Sham orbital. At DFT level wave functions are not well-defined, only the electron density. Then, the authors seem to show the LUMO to correlate it with the negative regions of the MEP. While it is true that this correlation works for the C atom, it does not for the icosahedral cage and the bottom B-H group. Despite the LUMO is as well localised in them, MEP does not show negative values in these regions. For the sake of clarity, I would suggest removing the analysis of the LUMO.
- Page 7, Line 314. It is written ... the eigenvalues of the later being .... The authors refer to the eigenvalues of the Laplacian of the electron density. There is not such a thing. They mean the eigenvalues of the Hessian matrix of the electron density.
- Page 9; Line 410. It is written NCH, CO2, FH, SiH2 and N2 ; in these systems the closed-shell interactions are important. The analysis of the Laplacian of the electron density reveals that the interaction in these systems is closed-shell. However in line 248 (page 6), it is said that the inversion in the order of the ΔG and ΔH for ClH and N2 is due to the noncovalent and covalent character. This is unclear. In addition, the authors do not comment on the role of entropy. The inversion may be more related to entropic factors than to the nature of the chemical bonding.
- Page 9; Line 415. It is written ... the potential energy V shows the effect of the electric field in the intermolecular region. This is not accurate. The local potential energy density or local virial field V is a measure of the average effective potential field experienced by a single electron in a many-particle system. The term electric field used by the authors may lead to confusion.
- Page 10; Line 420. The local energy and the Laplacian of the electron density have been widely used to characterised chemical interactions. However, no correlation is found between the values of ΔH, and those found in Fig 6b and Fig 6c. For instance, N2 shows a weak interaction as read from the value of ΔH. However, the electron density and the local energy density at the BCP are among the highest. For F- it is found just the opposite trend. By contrast, for CO2 and FH, there is a strong correlation. In line with the authors' claim in line 398, analysis of the electron density at BCPs are reasonable for the same pair of atoms. Otherwise misleading trends can be found. The authors may consider adding some words explaining this.
- Page 10; Line 443. It is said For example if a V(B,B) basin has a population of 0.8 electrons, the latter is not included in the group of the V(B,B). What the authors want to explain is not clear. Do they mean that if a basin V(B,B') has a population of 0.8 electrons, it is not attached to the superbasin formed by all the other V(B,B) basins?
- Page 10; Line 437. Table 2. I recommend including the information of the basins in the Supporting Information, Table S12.
- Page 11; Line 447. I suggest including the basins label in the caption of ELF pictures included in Table 2.
- Page 11; Lines 475-483. For the sake of clarity, I recommend inserting these lines at the beginning of Section 3.3.2. Otherwise, it might be confusing for the reader.
- Page 11; Line 490. ... the ELF does not localize a basin between the C(ylide) and the 0=C=0 molecule, and thus no bonding is expected neither a complex formation. This statement is quite strong. The absence of ELF basins indicates that electron pairs are not shared, and therefore, the interaction is not covalent. However, other interactions such as ionic or non-covalent interactions are possible even in the absence of ELF basins.
- Page 13; Conclusions. The main conclusions of the article are poorly written. In the first line, the authors claim that by means of quantum chemistry computations they can predict the formation of tetrel complexes. The power of quantum chemistry to predict free energies of formation is widely known. In addition, the verb predict is quite strong unless experimental results are not shown. In line 538, the authors validate their results with several structures found in the Cambridge structure database. This is quite positive, but, in my opinion, is not a prediction until experimental evidence is found. The strength of the article is the large versatility of CB11H11- to form tetrel bonds at room temperature.
Spelling mistakes
The article is chiefly written in British English. However, some words are written in American English. For the sake of coherence, I suggest correcting them:
- Page 7; Line 293: color (Am.) -- > colour (Br.)
- Page 7: Line 312: analyzing(Am.) --> analysing (Br.)
- Page 7; Line 313: analyzed (Am.) --> analysed (Br.)
Author Response
C: Comment from Reviewer, R: Reply
C1: Page 3; Line 100. The authors do not cite the code used to perform the ELF analysis. The references relative to ELF, 21 and 22, are not correct. They refer to the local analysis of the electron density.
R1: we have added the sentence:
"The TopMod09 package [23] was used for the ELF calculations."
at the end of the paragraph with ref [23]:
[23] Noury, S.; Krokidis, X.; Fuster, F.; Silvi, B. TopMod package, 1997.
and we have changed references 21 and 22 to:
[21] Silvi, B.; Savin, A. Nature 1994, 371, 683–686.
[22] Becke, A. D. ; Edgecombe, K. E. A simple measure of electron localization in atomic and molecular systems. J. Chem. Phys. 1990, 92, 5397–5403.
C2: Page 3; Line 102. The definition of ELF is wrong. ELF is connected to the probability of finding two electrons with the same spin.
R2: we have substituted the sentence:
"The ELF is a distribution function which measures the probability of finding an electron near to another electron with an opposite spin.."
by
"The ELF is a distribution function which measures the probability of finding two electrons with the same spin"
C3: Page 7; Line 271. It is written ...
the LUMO of the carbonium ylidehas considerable wave function amplitude around the C atom ...
Strictly speaking, for a many-electron system the LUMO is not a wave function. Additionally, this sentence is referred to as a Kohn-Sham orbital. At DFT level wave functions are not well-defined, only the electron density. Then, the authors seem to show the LUMO to correlate it with the negative regions of the MEP. While it is true that this correlation works for the C atom, it does not for the icosahedral cage and the bottom B-H group. Despite the LUMO is as well localised in them, MEP does not show negative values in these regions. For the sake of clarity, I would suggest removing the analysis of the LUMO.
R3: We have removed the analysis of the LUMO and rephrased the paragraph as follows:
"In Figure 5 we show for (1) the molecular electrostatic potential (MEP) and the electron localization function (ELF). These electronic structure features are computed using the optimized geometry of the system with the G4MP2 method – B3LYP/6-31G(2df,p) model chemistry for structure optimization. As noticed in Figure 5a, the shape of the MEP and the corresponding π-hole just on top of the C ylide center shows the electron-attraction nature of this region of the molecule. In the ELF from Figure 5b we show disynaptic V(B,H) yellow basins corresponding to the the B-H bonds; the ELF distribution around the CB11 icosahedral cage can be partitioned into green disynaptic and trisynaptic basins as we will described below in Section 3.3.2."
C4: Page 7, Line 314. It is written... the eigenvalues of the later being...
The authors refer to the eigenvalues of the Laplacian of the electron density. There is not such a thing. They mean the eigenvalues of the Hessian matrix of the electron density.
R4: we have rephrased the sentence as:
"...and Laplacian nabla^2(rho), with the eigenvalues of the Hessian matrix of the electron density being (l1, l2, l3)."
C5: Page 9; Line 410. It is written NCH, CO, FH, SiH and N; in these systems the closed-shell interactions are important. The analysis of the Laplacian of the electron density reveals that the interaction in these systems is closed-shell. However in line 248 (page 6), it is said that the inversion in the order of the ΔG and ΔH for ClH and N is due to the noncovalent and covalent character.This is unclear. In addition, the authors do not comment on the role of entropy. The inversion may be more related to entropic factors than to the nature of the chemical bonding.
R5: We have substituted the sentence from page 9, line 410:
"NCH, CO, FH, SiH and N; in these systems the closed-shell interactions are important."
to
"NCH, CO, FH, SiH and N; the analysis of the Laplacian of the electron density reveals that the interaction in these systems is closed-shell."
and on page 6, line 248, we have removed the sentence:
"this is due to the noncovalent and covalent character of each interaction"
C6: Page 9; Line 415. It is written ... the potential energy V shows theeffect of the electric field in the intermolecular region. This is not accurate. The local potential energy density or local virial field V is a measure of the average effective potential field experienced by a single electron in a many-particle system. The term electric field used by the authors may lead to confusion.
R6: We have substituted the sentence: "...the potential energy V shows the effect of the electric field in the intermolecular region."
by
"The local potential energy density or local virial field V is a measure of the average effective potential field experienced by a single electron in a many-particle system."
C7: Page 10; Line 420. The local energy and the Laplacian of the electron density have been widely used to characterised chemical interactions. However, no correlation is found between the valuesof ΔH, and those found in Fig 6b and Fig 6c. For instance, N shows a weak interaction as read from the value of ΔH. However,the electron density and the local energy density at the BCP are among the highest. For F it is found just the opposite trend. By contrast, for CO
and FH, there is a strong correlation. In line with the authors' claim in line 398, analysis of the electron density at BCPs are reasonable for the same pair of atoms. Otherwise misleading trends can be found. The authors may consider adding some words explaining this.
R7: The Reviewer is right: the topological analysis of BCP can describe the type of interaction but not the force/energy of the interaction. Therefore we have removed the word "attractive" in:
"Thus, H can be used for characterising the covalent character of an attractive interaction betweens atoms..."
and now the sentence reads:
"Thus, H can be used for characterising the covalent character of an interaction betweens atoms..."
and also have added at the end of the paragraph:
"Hence, the values of rho, nabla^2(rho) and H at BCP can describe the type of interaction, but not the attractive force or energy of interaction."
we have also removed the word "attractive" on Page 3, line 127.
C8: Page 10; Line 443. It is said For example if a V(B,B) basin has a population of 0.8 electrons, the latter is not included in the group of the V(B,B). What the authors want to explain is not clear. Do they mean that if a basin V(B,B') has a population of 0.8 electrons, it is not attached to the superbasin formed by all the other V(B,B) basins?
R8: We have substituted on Page 10; Line 443, the sentences:
"Below each ELF function of a given complex (1:n), we also report the function value and average population for the different types of basins. A threshold of 0.2 electrons is considered to include or not a basin in a group. For example if a V(B,B) basin has a population of 0.8 electrons, the latter is not included in the group of the V(B,B) basins with a population around 1.5."
by the following ones:
" Below each ELF function of a given complex (1:n), we report the population of the different basins. In order to avoid dealing with a long list of populations, and after observing that basins involving the same elements have similar populations, we decided to report only, for each type of basins, the average population. A threshold of 0.2 electrons was chosen to decide if two basins belong to the same group or not. For instance, if we consider basins V(B1,B2) and V(B1,B3) with populations of 0.8 and 1.5 electrons respectively, they belong to different groups. "
C9: Page 10; Line 437. Table 2. I recommend including the information of the basins in the Supporting Information, Table S12.
R9: We have selected four complexes with different d(C···X) distances for comparative purposes and believe that inclusion of the information of basins is important for the Reader. The ELF information for remaining complexes is compiled in the Supplementary Information file in order to complement the results for the Reader.
C10: Page 11; Line 447. I suggest including the basins label in the caption of ELF pictures included in Table 2.
R10: We have include the basins labels and the sentence in the Table caption:
"Basin labels are depicted for each complex."
C11: Page 11; Lines 475-483. For the sake of clarity, I recommend inserting these lines at the beginning of Section 3.3.2. Otherwise,it might be confusing for the reader.
R11: We have moved these lines at the beginning of Section 3.3.2
C12: Page 11; Line 490.... the ELF does not localize a basin betweenthe C(ylide) and the 0=C=0 molecule, and thus no bonding is expected neither a complex formation. This statement is quite strong. The absence of ELF basins indicates that electron pairs are not shared, and therefore, the interaction is not covalent. However,other interactions such as ionic or non-covalent interactions are possible even in the absence of ELF basins.
R12: We have substituted the sentence: "As regard to the (1:13) complex with CO2, the ELF does not localize a basin between the C(ylide) and the O=C=O molecule, and thus no bonding is expected neither a complex formation."
by the comments suggested by the Reviewer:
"As regard to the (1:13) complex with CO2, the ELF does not localize a basin between the C(ylide) and the O=C=O molecule; the absence of ELF basins indicate that electron pairs are not shared, and therefore, the interaction is not covalent. However, other interactions such as ionic or non-covalent are possible even in the absence of ELF basins."
C13: Page 13; Conclusions. The main conclusions of the article are poorly written. In the first line, the authors claim that by means of quantum chemistry computations they can predict the formation of tetrel complexes. The power of quantum chemistry to predict free energies of formation is widely known. In addition, the verb predict is quite strong unless experimental results are not shown. In line 538, the authors validate their results with several structures found in the Cambridge structure database. This is quite positive, but, in my opinion, is not a prediction until experimental evidence is found. The strength of the article is the large versatility of CB11H11(-) to form tetrel bonds at room temperature.
R13: We have changed
Page 13 line 552,
"we can predict the formation of tetrel complexes..."
by
"we should expect the formation of tetrel complexes..."
lines 556-557
"and we predict that all complexes can be formed"
by
"and the results indicate that all complexes should be formed..."
As regards to the Reviewer's comments on the CSD, at the end of the 1st Paragraph on page 13, we have rephrased the sentence by removing the statement "...hence there is a good agreement with experiment" as follows:
"Our (1:7) tetrel complex H11B11Cï€ ï‚¬:NH3 has a predicted d(C···N) = 1.498 Å, according to the G4MP2 computational model."
C14: The article is chiefly written in British English. However, some words are written in American English. For the sake of coherence, I suggest correcting them:
Page 7; Line 293: color (Am.) -- > colour (Br.)
Page 7: Line 312: analyzing(Am.) --> analysing (Br.)
Page 7; Line 313: analyzed (Am.) --> analysed (Br.)
The three spelling mistakes are corrected in the revised manuscript according to British English.
Reviewer 2 Report
The article "Tetrel bonding in CB11H11 carbonium ylide" by Maxime Ferrer, Ibon Alkorta, José Elguero, and Josep M. Oliva-Enrich presents computational studies on intermolecular interactions of CB11H11 molecule and several Lewis bases. Although the article is interesting and the calculations performed are valuable, in my opinion, the described material does not fit into the scope of the Crystals. The only link I can find is the study of intermolecular interactions. However, in the case of the presented work, they are not tested in the crystalline material. In my opinion, the work should be sent to a more theoretical journal.
Author Response
C: Comments from the Reviewer, R: Reply to the Reviewer
C: The article "Tetrel bonding in CB11H11 carbonium ylide" by Maxime Ferrer, Ibon Alkorta, José Elguero, and Josep M. Oliva-Enrich presents computational studies on intermolecular interactions of CB11H11 molecule and several Lewis bases. Although the article is interesting and the calculations performed are valuable, in my opinion, the described material does not fit into the scope of the Crystals. The only link I can find is the study of intermolecular interactions. However, in the case of the presented work, they are not tested in the crystalline material. In my opinion,the work should be sent to a more theoretical journal.
R: We appreciate the Reviewer's comments:
However, the work consists of the study of electronic interactions that take place between carbonium ylide CB11H11 and Lewis bases, with experimental crystal data on several examples ( Refs. 33-39 and Refs. ).
Reviewer 3 Report
Maxime Ferrer et al reported a research article on “Carboranes as Lewis acids: Tetrel bonding in CB11H11 carbonium ylide”. Fourth generation (G4MP2) quantum chemical calculations were performed to study the electronic interaction between carbon in the carbenoid (1) and Lewis base in the complexes involving tetrel bonds. The geometries of the complexes were optimized at B3LYP/6-31G(2df,p) level of theory followed by single point energy calculations and frequency calculations to make sure local minimum is achieved for the optimized structures. The free energy of formation (ΔG) and enthalpy (ΔH) for complexes were calculated to know which complexes have the ability to spontaneously form at room temperature. Electronic structures of the complexes were analyzed using LUMO, electrostatic potential, electron localization function. Topological analysis of the complexes were carried out using Quantum Theory of Atoms in Molecules (QTAIM).
This research article will attract interdisciplinary readers who are interested in computational chemistry, Boron chemistry. This manuscript fit the aim and scope of the journal “Crystals”. Authors presented results clearly and discussed thoroughly. Therefore, reviewer recommends considering this manuscript for publication.
Comments:
- It is a good idea to provide supporting information to the peer reviewers at the time of review to gain more understanding and to provide constructive and critical feed back to the authors so that the article will be improved further.
- Authors are suggested to go through the paper more carefully and check for spellings.
(For example: line 496 “depens, should be depends”
(For example: line 546 “ fluor should be fluorine”
Author Response
1. It is a good idea to provide supporting information to the peer reviewers at the time of review to gain more understanding and to provide constructive and critical feed back to the authors so that the article will be improved further.
Reply: We have indeed provided a supporting information file at the time of submission of this work.
2. Authors are suggested to go through the paper more carefullyand check for spellings.
For example: line 496 “depens," should be "depends"
For example: line 546 “fluor" should be "fluorine”
Reply: we have corrected the spellings on lines 496 and 546.
Round 2
Reviewer 1 Report
The authors have implemented all the suggested amendments and the article has been dramatically improved. Nevertheless, it seems that some of the amendments have been made careless.
Figure 1 has not been corrected. The authors have removed the analysis of the LUMO from the main text, but not from Figure 1 and its caption.
Page 8; Line 317. It is written ... ..and Laplacian nabla^2(rho), with the eigenvalues of the Hessian matrix of the electron density being (l1, l2, l3). As suggested, the authors have corrected the original sentence and now it is written that λ1, λ2, λ3 are the eigenvalues of the Hessian matrix of the electron density. What the authors have not done, is check the syntax of the sentence. The authors may want to say: The topological properties of ρ(r) are analysed with the gradient of ∇ρ(r), the Laplacian of ρ(r), ∇2ρ(r), and the eigenvalues of the Hessian matrix of the electron density λ1, λ2, λ3.
Page 9; Line 411: It is written ... the shared interactions for ∇2ρ(r) > 0 .... Shared-interactions are characterised for having ∇2ρ(r) < 0.
Page 10; Line 421: It is written ... the potential energy V shows the effect of the electric field in the intermolecular region. Despite the authors' answer, this statement has not been corrected in the new version of the manuscript.
Page 10; Line 426: It is written: Thus, H can be used for charactersing the covalent character of an interaction between atoms [28]. and a combination of the Laplacian and H for the characterisation of hydrogen bond [29]. Hence, the values of ρ, ∇2ρ and H at BCP can describe the type of interaction, but no the attractive force or energy of interaction. The authors may assume that from the previous comments on Figure 6c, it is clear the ρ, ∇2ρ and H, can be used as bond descriptors. The strengths and limitations of these bond descriptors are already known and not a result of the manuscript. For the sake of clarity, I would suggest removing the aforementioned paragraph.
There are still some spelling mistakes:
Page 1; Line 15: centeres -->centres.
Page 7; Line 279: we will described --> describe.
Author Response
1) Figure 1 has not been corrected. The authors have removed the analysis of the LUMO from the main text, but not from Figure 1 and its caption.
Answer: We have corrected Figure 1 and its caption accordingly.
2) Page 8; Line 317. It is written ... ..and Laplacian nabla^2(rho), with the eigenvalues of the Hessian matrix of the electron density being (l1, l2, l3). As suggested, the authors have corrected the original sentence and now it is written that λ1, λ2, λ3 are the eigenvalues of the Hessian matrix of the electron density. What the authors have not done, is check the syntax of the sentence. The authors may want to say: The topological properties of ρ(r) are analysed with the gradient of ∇ρ(r), the Laplacian of ρ(r), ∇2ρ(r), and the eigenvalues of the Hessian matrix of the electron density λ1, λ2, λ3.
Answer: We have included the sentence suggested by the Reviewer.
3) Page 9; Line 411: It is written ... the shared interactions for ∇2ρ(r) > 0 .... Shared-interactions are characterised for having ∇2ρ(r) < 0.
Answer: we have corrected the equation.
4) Page 10; Line 421: It is written ... the potential energy V shows the effect of the electric field in the intermolecular region. Despite the authors' answer, this statement has not been corrected in the new version of the manuscript.
Answer: We have substituted the sentence accordingly.
5) Page 10; Line 426: It is written: Thus, H can be used for charactersing the covalent character of an interaction between atoms [28]. and a combination of the Laplacian and H for the characterisation of hydrogen bond [29]. Hence, the values of ρ, ∇2ρ and H at BCP can describe the type of interaction, but no the attractive force or energy of interaction. The authors may assume that from the previous comments on Figure 6c, it is clear the ρ, ∇2ρ and H, can be used as bond descriptors. The strengths and limitations of these bond descriptors are already known and not a result of the manuscript. For the sake of clarity, I would suggest removing the aforementioned paragraph.
Answer: we have removed the paragraph.
6) There are still some spelling mistakes:
Page 1; Line 15: centeres -->centres.
Page 7; Line 279: we will described --> describe.
Answer: Both mistakes have been corrected.
Reviewer 2 Report
As in the first round, the article does not raise my doubts from the substantive point of view. After reading carefully Crystals' Aims & Scope (https://www.mdpi.com/journal/crystals/about), I do not see a place where this article might fit. However, the Editor confirmed that this type of article might be of the journal's interest. Since most of the flaws have already been fixed, I can recommend this article for publication. Editorial board should decide whether to publish this paper in Crystals or not.
Author Response
We appreciate the Reviewer's viewpoint.